

# Real-time multilingual speech recognition and speaker diarization system based on Whisper segmentation

Ke-Ming Lyu[1], Ren-yuan Lyu[1] and Hsien-Tsung Chang[1,2,3]

[1] Computer Science and Information Engineering, Chang Gung University, Taoyuan, Taiwan
[2] Physical Medicine and Rehabilitation, Chang Gung Memorial Hospital, Taoyuan, Taiwan
[3] Bachelor Program in Artificial Intelligence, Chang Gung University, Taoyuan, Taiwan

## ABSTRACT

This research presents the development of a cutting-edge real-time multilingual speech recognition and speaker diarization system that leverages OpenAI's Whisper model. The system specifically addresses the challenges of automatic speech recognition (ASR) and speaker diarization (SD) in dynamic, multispeaker environments, with a focus on accurately processing Mandarin speech with Taiwanese accents and managing frequent speaker switches. Traditional speech recognition systems often fall short in such complex multilingual and multispeaker contexts, particularly in SD. This study, therefore, integrates advanced speech recognition with speaker diarization techniques optimized for real-time applications. These optimizations include handling model outputs efficiently and incorporating speaker embedding technology. The system was evaluated using data from Taiwanese talk shows and political commentary programs, featuring 46 diverse speakers. The results showed a promising word diarization error rate (WDER) of 2.68% in two-speaker scenarios and 11.65% in three-speaker scenarios, with an overall WDER of 6.96%. This performance is comparable to that of non-real-time baseline models, highlighting the system's ability to adapt to various complex conversational dynamics, a significant advancement in the field of real-time multilingual speech processing.

## INTRODUCTION

With the rapid advancement of artificial intelligence technology, automatic speech recognition (ASR) has become an indispensable part of our daily lives. The applications of speech recognition are extensive, ranging from voice search, voice command, and video subtitling to personal assistants in smart devices, pervading every aspect of life. However, traditional speech recognition systems have focused primarily on processing a single language and often perform inadequately in multispeaker scenarios, which limits their practicality and accuracy.

In September 2022, OpenAI introduced a significant breakthrough in speech recognition technology with the launch of the Whisper model (*Radford et al., 2023*). This

Corresponding author
Hsien-Tsung Chang,
smallpig@widelab.org

model, based on the Transformer encoder-decoder architecture, leverages a vast amount of weakly supervised data from multiple languages, encompassing a variety of accents and intonations. The rich training dataset endows the Whisper model with exceptional generalization capabilities, enabling it to maintain efficient recognition performance across different languages and in multispeaker environments.

In addition to speech recognition, speaker diarization plays a pivotal role in the field of speech processing, primarily addressing the question of who speaks when. This involves analyzing the voice characteristics of different individuals, segmenting the original audio signal, and categorizing it based on each person's unique features. Unlike speaker identification, which identifies specific individuals, speaker diarization merely distinguishes between different speakers without requiring any prior information about them, making it more widely applicable in real-world scenarios. Speaker diarization is applicable in the analysis of various audio data, such as media broadcasts, conference conversations, online social media, and court proceedings, and it also helps enhance the accuracy of speech recognition in multispeaker settings.

In recent years, research integrating automatic speech recognition and speaker dimerization techniques has begun to emerge. By combining these technologies, it is possible not only to accurately recognize speech content but also to determine when different individuals are speaking, thereby resolving the issue of who said what and when. The applications of this technology are broad, encompassing multiparty meeting transcription, analysis of multispeaker broadcast programs, and the construction of court records, among other multiparty conversational tasks, significantly improving the accuracy and practicality of speech processing in these applications.

Another critical area that warrants attention is the demands and challenges associated with real-time systems. In today's diverse era of digital communication, there is a growing need for rapid and effective processing and response to voice data, especially in scenarios involving multiple participants. Real-time voice processing systems, such as real-time speech recognition and speaker diarization systems, are capable of processing and analyzing speech as it occurs. This has significant implications for enhancing meeting efficiency, improving customer service experiences, and strengthening the interactivity of multimedia content. The real-time processing capabilities of these systems not only improve communication efficiency but also enable participants to interact and respond more immediately.

Against this backdrop, the development of real-time speech recognition combined with speaker diarization systems has become crucial. This technology is particularly suited for scenarios requiring immediate responses, such as real-time captioning for multiparty meetings and live broadcasts. However, integrating automatic speech recognition with speaker diarization in real-time systems faces numerous challenges. In terms of speech recognition, the system must generate recognition results immediately upon receiving speech; thus, it can only process a small segment of speech at a time. This processing approach may lead to a decrease in overall recognition accuracy. Regarding speaker diarization models, the inability to process the entire speech stream at once can result in

```
"id": 0,
"seek": 0,
"start": 0.0,
"end": 2.5,
"text": "我覺得多少還是會有吧",
"tokens": [ 50364, 17689, 41492, 25583, 6236, 2412, 6062, 50489],
"temperature": 0.0,
"avg_logprob": -0.33182077026367185,
"compression_ratio": 1.4941176470588236,
"no_speech_prob": 0.026732061058282852
```

**Figure 1 Segments of the whisper model.**     

the misallocation of speech segments to the wrong speakers during the clustering process, potentially leading to local optimum solutions rather than global ones, causing classification confusion and affecting clustering outcomes. Therefore, the key to effective real-time systems lies in how accurately the speech stream can be segmented to ensure that each speech segment is correctly attributed to a specific speaker while also maintaining the accuracy of speech content recognition. Moreover, considering the low-latency characteristics of real-time systems, finding an appropriate balance between accuracy and processing efficiency is essential.

Notably, the text segments identified by the Whisper model, as illustrated in Fig. 1, typically contain speech from only one individual. Even if the speech of different individuals overlaps in time, the model effectively segments it into distinct pieces. This discovery provides a crucial foundation for our system, enabling it to directly utilize the speech segments segmented by the model for speaker identification and labeling. This obviates the need for the subtasks mentioned earlier, such as speaker change detection and speaker resegmentation, reducing the complexity of subsequent speaker diarization tasks and further reducing the latency of real-time systems.

The objective of this study is to leverage the characteristics of Whisper's speech recognition output, where each segment contains speech from only one individual, to simplify the complex processes of traditional speaker diarization methods. Based on this foundation, we developed a real-time multilanguage speech recognition and speaker diarization system that addresses the issue of achieving local optimal solutions in real-time voice signals rather than global optimal solutions. This approach increases the accuracy to a level comparable with that of other non-real-time speech recognition and speaker diarization systems while also reducing the latency of real-time systems, enhancing their practical value in various scenarios.

In summary, the principal contributions of our research are twofold: (1) We discovered and utilized the characteristic that each segment identified by Whisper speech recognition contains only one speaker, which simplifies the complex processes associated with traditional speaker diarization methods to a certain extent. (2) Building upon this discovery, we adjusted the incremental clustering method in the foundational work of *Coria et al. (2021)* and developed a real-time multilanguage speech recognition and speaker labeling system. This system effectively addresses the issue of local optimal solutions in real-time voice streams, increasing the accuracy to levels comparable to those

of other non-real-time ASR+SD systems. Moreover, it significantly enhances the efficiency of speaker diarization and speech recognition in real-time systems in multilanguage and multispeaker environments, thereby offering immense practical value for real-world applications.

## RELATED WORKS

In the literature review section, we explore four main topics: end-to-end speech recognition, speaker diarization, the combination of speech recognition and speaker diarization, and real-time speaker diarization systems.

### End-to-end speech recognition

End-to-end (E2E) speech recognition involves the integration of acoustic models, alignment mechanisms, and language models into a single large-scale model. This approach simplifies the system architecture by reducing the complex interactions and error accumulation between different components. It learns directly from raw audio data, avoiding the tedious feature engineering process and offering stronger adaptability to different languages and accents. Early end-to-end modeling methods explicitly modeled alignment through a latent variable, which was marginalized out during training and inference. For instance, *Graves et al. (2006)* introduced the connectionist temporal classification (CTC) model, which addresses the alignment between input and output sequences of varying lengths by introducing a "blank" label, allowing the model to learn how to map acoustic features to text sequences. One significant advantage of the CTC model is its ability to learn directly from sequences of speech data to text labels without the need for manually annotated alignment, thereby simplifying the entire learning process. Furthermore, in 2012, Graves (*Graves, 2012*) proposed the recurrent neural network transducer (RNN-T) model, which models acoustic features with an RNN and uses a transducer to directly generate text output. The RNN-T model retains CTC's advantages while enhancing the modeling capacity for more complex alignments between speech and text sequences, making it more effective for processing natural and fluent speech data. However, these models also have limitations. For example, the CTC model may experience a decrease in accuracy when dealing with speech that has significant variations in pace or dialect accents, while the RNN-T model, despite offering better alignment capabilities, has a higher computational complexity, which can lead to slower processing.

Between 2016 and 2020, end-to-end (E2E) automatic speech recognition models based on attention mechanisms, such as the Listen, Attend and Spell (LAS) model introduced by *Chan et al. (2016)*, made significant strides in the field of speech recognition. The LAS model is a neural network that combines listening, attending, and spelling functions to directly convert speech into characters, eliminating the need for pronunciation models, hidden Markov models, or other components typically used in traditional speech recognition systems. The LAS model utilizes an attention mechanism to implicitly identify and model the relevant parts of the input acoustics, conditionally modeling both the entire acoustic sequence and the previously generated characters, thereby producing each

character individually. This approach not only improves the processing of long texts but also achieves more accurate recognition in complex contexts.

Furthermore, following the breakthrough of the Transformer model (*Vaswani et al., 2017*) in sequence-to-sequence tasks in 2017, various Transformer-based speech recognition models emerged, such as *Dong, Xu & Xu (2018)* and *Zhang et al. (2020)*. These models employ multilayer Transformers based on self-attention to individually encode audio and text label sequences. They also use a cross-attention mechanism to calculate the associations between the speech at each time point and the previously recognized text, computing the probability distribution of the next text label. This process does not solely rely on the audio signal but also considers the context to predict the next possible text. This method enhances the understanding of the information in the speech stream, significantly improving recognition rates for lengthy speech segments.

## Speaker diarization

Speaker diarization models, such as *Bredin et al. (2020)*, encompass a series of methods, including voice activity detection (VAD), speech segmentation, feature extraction, clustering, and resegmentation. The purpose of VAD is to extract speech signals in which speech activity is present from raw audio, while speech segmentation aims to cut the audio into smaller segments based on speaker change points, ensuring that each segment acoustically belongs to the same individual. Feature extraction techniques such as i-vectors (*Dehak et al., 2009*), d-vectors (*Variani et al., 2014*), and x-vectors (*Snyder et al., 2018*) utilize Gaussian mixture models (GMMs) (*Reynolds & Rose, 1995*), deep neural networks (DNNs), and other technologies to extract acoustic features that characterize speakers. During the clustering phase, these acoustic features are used to classify scattered speech segments, assigning each individual to a distinct category and outputting speaker labels for each segment. Resegmentation is employed to refine and adjust the boundaries of labeled speaker segments using more detailed speech features and models to enhance the accuracy of speaker diarization.

However, in these traditional speaker diarization methods, the VAD and speech segmentation stages are susceptible to background noise and sound quality issues, leading to inaccurate detection of speech activity. Furthermore, the feature extraction and clustering steps rely on manually designed acoustic features and conventional clustering algorithms, limiting the system's adaptability and accuracy across different speakers and complex acoustic environments. Additionally, these methods struggle to effectively address the issue of speaker overlap.

## Combining speech recognition with speaker diarization

In the past 1 to 2 years, there has been a growing trend toward integrating speech recognition with speaker diarization techniques to enhance the practicality of both speech recognition and speaker diarization. For example, *El Shafey, Soltau & Shafran (2019)* focused on integrating speaker labels into end-to-end speech recognition models. They proposed adding speaker labels with specific roles, such as "doctor" and "patient," to the output of a speech recognition system based on recurrent neural network

transducers (RNN-T). This method was validated in medical dialogs, significantly reducing the word diarization error rate (WDER) while minimally impacting the overall word error rate (WER). This research demonstrates that integrating speaker labels into speech recognition outputs effectively facilitates the concurrent execution of speech recognition and speaker diarization, offering a direct and promising solution. However, this approach requires determining and fixing speaker roles or identity labels during the training stage, which may pose challenges when dealing with an undetermined number of speakers.

Following the release of the Whisper model in September 2022, *Bain et al. (2023a)* and *Bain et al. (2023b)* utilized the Whisper model (*Radford et al., 2023*) in combination with an end-to-end ensemble multiclass classification speaker diarization model (*Plaquet & Bredin, 2023*). They processed the same segment of speech through both speech recognition and speaker diarization models, generating a text containing multiple temporal segments and a sequence of speaker labels with multiple temporal segments. By calculating the intersections on the temporal axes of both, they aligned the text segments with the speaker labels. This method ensures that sentences containing only one speaker are not mistakenly assigned, significantly improving the accuracy of both speech recognition and speaker diarization. This model is also used as the benchmark for the final comparison.

### Real-time speaker diarization

*Coria et al. (2021)* introduced a low-latency real-time speaker diarization system, which is fundamentally viewed as a combination of incremental clustering and local diarization. The system utilizes a rolling buffer that updates every 500 milliseconds and employs a specially designed end-to-end overlap-aware segmentation model to detect and separate overlapping speech. Additionally, the system includes a modified statistics pooling layer, which assigns lower weights to frames predicted to have simultaneous speakers. A key feature of the system's design is its initial handling of overlapping speech, which contrasts with traditional methods that treat overlapping speech processing as a postprocessing step. This approach allows the system to excel in achieving low latency and high accuracy. Our research also further modifies the architecture of this real-time system, integrating it with the Whisper speech recognition model to establish a more comprehensive low-latency system.

## METHODS

In the experimental methodology section, we detail the techniques and methods employed in this study, including the way we handle continuous speech streams and integrate ASR with SD technologies. We begin with an overview of the system's architecture, explaining its working principles and the functions of its key components. We then delve into the specific technical implementations, covering critical aspects such as speech recognition, speech segmentation, speaker embedding, and incremental clustering. We also elucidate the differences between real-time and non-real-time system methodologies.

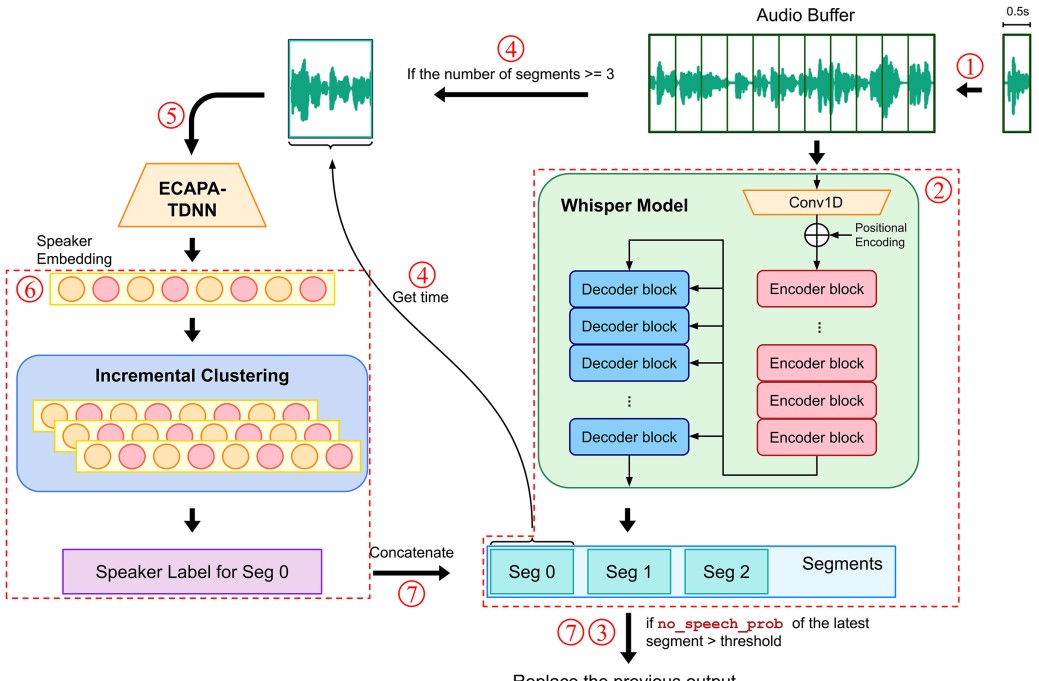

**Figure 2  Real-time automatic speech recognition and speaker diarization workflow.**

## Overview

The overall operation flow of our real-time system is illustrated in Fig. 2. The following explanation proceeds in the order of the numbered steps in the flowchart:

1. The system extracts audio blocks every 0.5 s from a continuous speech signal and processes them in these fixed time segments. This approach ensures the continuity and completeness of the data, avoiding overlap and loss of audio information. These audio blocks are then stored in an audio buffer for subsequent processing.

2. The Whisper model is used to perform speech recognition on all audio data in the buffer, automatically generating several corresponding speech segments.

3. The recognition results are displayed on the screen. At this stage, the output has not yet been assigned speaker labels and is dynamically updated based on whether the probability of no speech (no_speech_prob) in the latest processed segment exceeds a preset threshold. If so, the segment is deemed silent, and the previous output is retained; otherwise, the previous output is overwritten. This mechanism ensures that the information displayed on the screen is timely and accurate while excluding nonspeech noise, thereby achieving real-time processing and updating of speech data in a multilingual environment.

4. When the number of speech segments output by the Whisper model reaches or exceeds a certain quantity, the system extracts an audio signal of the same length as the first segment from the buffer, treating this segment as containing speech from only one speaker.

5. A speaker embedding model is used to extract the acoustic features of the speaker, producing speaker embeddings.

6. These speaker embeddings are then subjected to an incremental clustering algorithm, which clusters them with previously inputted speaker embeddings to instantly identify and generate corresponding speaker labels.

7. The speaker labels are combined with the previously generated speech recognition text segments from the Whisper model and output to the screen. At this point, the output, which includes speaker labels and recognized text, is considered the final result and is retained on the screen, not to be overwritten by subsequent recognition results.

## Speech recognition and speech segmentation

In our study, we employed the speech recognition model Whisper, proposed by OpenAI (*Radford et al., 2023*), as the foundation. The Whisper model adopts an encoder-decoder transformer architecture trained on a vast dataset of weakly labeled and pseudolabeled audio data, which is capable of recognizing up to 98 languages. In multilingual tasks, the Whisper model, through training on diverse audio conditions and a wide range of language datasets, has an enhanced ability to recognize different languages and accents. Moreover, the model utilizes specific data preprocessing and feature normalization techniques, as well as a specially designed text tokenizer for various languages, to improve its performance in multilingual speech recognition tasks. The design of this multilingual version enables the Whisper model to be widely applied globally and to handle speech recognition tasks in various linguistic environments.

The Whisper model incorporates a small initial module containing two convolutional layers, where the stride of the second convolutional layer is 2, and uses filters of width 3 with the GELU activation function. After processing by these convolutional layers, sinusoidal positional embeddings are added to the output of the initial module, followed by the application of encoder Transformer blocks.

In our research, we delve into the segmentation proposed by the Whisper model, focusing on its contribution to subsequent speaker identification efforts. We find that although Whisper does not directly recognize different speakers, it can effectively segment the speech of different speakers into independent sentences. This approach is particularly valuable for processing multispeaker dialog scenarios, as Whisper can accurately segment the speech of each speaker, even when they are speaking simultaneously. This segmentation capability provides subsequent speaker embedding models with clearly separated speech data, significantly improving the accuracy and efficiency of speaker identification and labeling. In other words, while Whisper's segmentation feature does not directly involve speaker identification, it provides crucial preliminary processing for accomplishing this task, which is indispensable for the entire speaker diarization and processing workflow.

However, in the application to real-time systems, we observed some uncertainty in the Whisper model regarding the number of speech output segments, meaning that the output segment count is not strictly incremental, and the distribution of these segments appears to

fluctuate. For example, during an ongoing speech stream, two segments might be output in one second, one in the next, and then four in the following second. We found that only when the segment count reaches a certain level does the timeline become sufficiently accurate. Therefore, in practical applications, it is necessary to set a reasonable segment count threshold to determine when to start subsequent processing while also considering the accuracy of the timeline and system latency issues.

## Speaker embedding

In our research, we utilized speaker embedding technology to extract the voice characteristics of speakers from the original audio signal segmented according to the timeline provided by the Whisper model. We selected the enhanced channel attention, propagation, and aggregation in TDNN (ECAPA-TDNN) model (*Desplanques, Thienpondt & Demuynck, 2020*) based on a time delay neural network (TDNN). This model represents an advanced speaker identification architecture that excels in capturing key features of speech data.

In the preprocessing and feature extraction stage, the ECAPA-TDNN model preprocesses the original audio signal, including converting the sound into 80-dimensional mel frequency cepstral coefficients (MFCCs), which is achieved by using a 25-millisecond window and a 10-millisecond step size. Cepstral mean subtraction is then applied to normalize these feature vectors.

The ECAPA-TDNN model employs several key techniques to enhance its performance. The first is channel- and context-dependent statistical pooling, which uses soft attention to calculate weighted statistics in the temporal pooling layer. By extending the attention mechanism to the channel dimension, the model can focus more on speaker characteristics activated at different time points. Additionally, ECAPA-TDNN introduces one-dimensional squeeze-excitation (SE) in residual blocks with two layers (Res2Blocks), generating descriptors for each channel by calculating the mean vector of frame-level features in the temporal domain and readjusting the frame-level features based on global speech attributes.

Finally, through multilayer feature aggregation and summation (MFA) technology, the model combines the output feature maps of all the SE-Res2Blocks, thereby enhancing the robustness and discriminability of the speaker embeddings. When the original audio signal is input into the ECAPA-TDNN model after this series of processes, it ultimately outputs a 192-dimensional vector that comprehensively reflects the unique voice characteristics of the speaker, providing a solid foundation for our speech recognition and speaker diarization system.

## Incremental clustering

The primary distinction between real-time speaker diarization systems and non-real-time systems lies in their ability to swiftly update clustering results as new speech data arrives, a process known as incremental clustering. The major advantage of this approach is its ability to process input vectors in real time, effectively addressing potential discontinuities

in identification that may arise during the clustering process, thereby ensuring consistency and accuracy in the clustering outcomes.

We adapted the method proposed by *Coria et al. (2021)* and made adjustments to this clustering workflow. According to the method of *Coria et al. (2021)*, each sliding window generates three local speaker labels based on the results of speaker segmentation. Each update maps these three labels to global speaker labels while ensuring that different local speaker labels within the same sliding window are not mapped to the same global speaker label. This mechanism maintains clear distinctions between speakers, facilitating accurate tracking and identification of speakers throughout the audio stream. This adjusted approach to incremental clustering is crucial for maintaining high performance in real-time diarization systems, particularly in environments where speaker overlaps and quick speaker changes are common.

Since we output only one speaker embedding at each time point, our method modifies the process to involve only one local speaker at a time, eliminating the need to consider the issue of different local speakers being mapped to the same global speaker label simultaneously. This modification simplifies the original incremental clustering method, reducing the computational complexity. Initially, based on a predefined maximum number of speakers $N$, the system initializes $N$ sets of 192-dimensional zero vectors as the starting points for all centroids, denoted as $C = \{c_1, c_2, \ldots, c_N\}$. For each incoming speaker embedding $e$, the system calculates the cosine distance $d$ between it and each centroid $c_i$ with the formula $d(e, c_i) = \frac{e \cdot c_i}{||e|| ||c_i||}$. A threshold $\delta_{\text{new}}$ is set to assess the similarity between the newly inputted speaker embedding $e$ and the existing centroids. When the distance between a centroid and $e$ is less than this threshold, they are considered sufficiently similar, and the centroid may need to be updated to reflect the characteristics of the new data.

If the current number of centroids $M$ equals the predefined maximum number $N$, the system cannot create new centroids. In this case, the system directly finds the centroid closest to $e$ for assignment. If the distance between this centroid and $e$ is greater than $\delta_{\text{new}}$, the centroid is not updated; otherwise, it is updated. This step can be expressed as follows:

$$\begin{cases} s = \underset{i \in C}{\arg\min}\, d(e, c_i), & \text{if } M = N \\ c_s \leftarrow c_s + e, & \text{if } M = N \text{ and } d(e, c_s) \leq \delta_{\text{new}} \end{cases} \tag{1}$$

Next, we calculate the set $m = \{i \mid d(e, c_i) < \delta_{\text{new}}\}$ to determine which centroids have a distance from $e$ smaller than the threshold $\delta_{\text{new}}$, and the next steps are taken based on the following condition:

$$\begin{cases} s = \underset{i \in m}{\arg\min}\, d(e, c_i), & \text{if } |m| > 0 \\ c_s \leftarrow c_s + e \\ c_{M+1} \leftarrow e, & \text{otherwise} \end{cases} \tag{2}$$

If the set $m$ is not empty, the number of the centroid $s$ in $m$ that is closest to $e$ is selected as the speaker label corresponding to the speaker embedding $e$, and the value of that centroid $c_s$ is updated to $c_s + e$. If $m$ is empty and $M$ is less than $N$, the system creates a new

centroid $c_{M+1}$ and sets it as $e$. This method ensures that the clustering results can be updated in real time and effectively handles new speaker embeddings when the number of centroids reaches the limit, maintaining the accuracy of clustering.

## EXPERIMENTS

In the experimental section, we elaborate on the methodologies employed in our study, including the methods of collecting our testing data, the evaluation metrics we utilized, the baseline models against which our system was compared, the settings of the experimental parameters, the methods of documenting the experimental results, and the testing of the latency of our real-time system. This detailed exposition aims to provide a clear and comprehensive understanding of how our system performs under various conditions and how it compares to existing standards in the field.

### Test data collection

In our study, we focus on evaluating the performance of our system in real-world application scenarios, especially in multilingual environments, Mandarin speech with Taiwanese accents, and situations where the speakers frequently alternate. To this end, we collected several Taiwanese interviews and political commentary programs from YouTube to serve as our evaluation data. These videos collectively feature 46 different individuals and have a total duration of 4 h, 9 min, and 19.48 s. Each video includes two to three people with different ages, genders, and accents. The languages used are predominantly Mandarin but also include some English and Taiwanese Hokkien. This diversity not only enriches our *corpus* but also means that the data more closely resemble real-world speech usage scenarios. Table 1 displays detailed statistical data on the testing dataset.

During the data processing phase, we used OpenAI's Whisper Large-v3 model for speech recognition. Afterward, manual checks and labeling were conducted to ensure the accuracy of the speech segments and speaker labels. In the proofreading process, we focused on maintaining the consistency and accuracy of speaker labels and the correctness of sentence arrangement without strictly correcting recognition errors in text or time axis alignment. This approach allowed us to effectively focus our resources on ensuring the accuracy of speaker labels, which is crucial for our research objectives.

In the data preprocessing stage before model evaluation, we implemented a series of standardization measures, including removing all punctuation marks, converting between traditional and simplified Chinese characters, and separating Chinese and English texts. In these standardization steps, Chinese characters and English words were considered the basic units of analysis to ensure the accuracy of subsequent analyses.

### Evaluation metrics

Traditional speaker diarization systems are typically evaluated using the diarization error rate (DER), which focuses on comparing preannotated speaker segments with the system's time-domain predictions. However, in systems combining ASR with SD, where speaker roles are directly assigned to identified words, relying on time boundaries to align words with speaker roles is unnecessary.

**Table 1 Testing data analysis.**

|  | Num. of videos | Total Spk. | M Spk. | F Spk. | Total Dur. | Avg. Dur. |
|---|---|---|---|---|---|---|
| Two-speakers | 12 | 24 | 15 | 9 | 2:06:25.83 | 10:32.15 |
| Three-speakers | 10 | 26 | 16 | 10 | 2:00:56.65 | 12:05.66 |
| Total | 24 | 46 | 29 | 17 | 4:06:22.48 | 10:18.44 |

**Note:**
Num. of videos, number of videos; Total Spk., total speakers; M Spk., male speakers; F Spk., female speakers; Total Dur., total duration; Avg. Dur., average duration.

In this context, *El Shafey, Soltau & Shafran (2019)* proposed a new evaluation method specifically designed for assessing end-to-end systems that combine speech recognition and speaker diarization. This method assesses performance by measuring the percentage of words correctly labeled with speaker tags in the transcribed text. Unlike the DER, this new evaluation standard, named the word diarization error rate (WDER), focuses on the proportion of 'words' labeled with the wrong speaker rather than the traditional 'time' proportion.

$$WDER = \frac{S_{IS} + C_{IS}}{S + C}.$$

Here, $S_{IS}$ represents the number of incorrectly labeled speaker tags in ASR substitutions, $C_{IS}$ is the number of incorrectly labeled speaker tags in correctly recognized ASR words, $S$ is the number of ASR substitutions, and $C$ is the number of correctly recognized ASR words. Given that ASR and SD systems focus on accurately assigning words to the corresponding speakers rather than just on time-alignment accuracy, we believe that the WDER is a more effective metric for reflecting the overall performance of the system, as it aligns more closely with the needs of practical application scenarios.

Additionally, *El Shafey, Soltau & Shafran (2019)* mentioned that since the WDER does not include calculations for insertions and deletions, it is advisable to use it in conjunction with the word error rate (WER) for a comprehensive evaluation of the entire speech recognition and speaker diarization system. Based on this, we also adopted the WER to evaluate our model, primarily as a supplementary metric to provide a comprehensive performance perspective. This is because the main focus of our system is to enhance the accuracy and efficiency of speaker diarization rather than the accuracy of the speech recognition results.

## Baseline

The baseline model used in our study combines the Whisper speech recognition system with Pyannote Speaker Diarization technologies (*Bain et al., 2023b*, *2023a*; *Plaquet & Bredin, 2023*; *Bredin, 2023*). This model processes the input speech simultaneously through the Whisper model and a pretrained speaker separation model, producing two sets of outputs: Whisper segments and speaker separation results. These outputs are then merged by calculating the intersection of each speech segment with the speaker time segments, and speakers are assigned based on the sizes of these intersections. Specifically, each speech or

text segment is assigned to the speaker with the largest time intersection, achieving optimal speaker alignment and allocation. The system aligns the text transcriptions generated by the automatic speech recognition system with the identified speaker time segments. This process involves calculating the intersections and unions between speech paragraphs and text transcriptions to determine their correlation. When a speech segment overlaps with a specific text transcription, that text is attributed to the corresponding speaker. Furthermore, even in the absence of direct overlap, the system can find and assign the closest speaker to each text segment through a certain mechanism. This alignment method ensures the accurate attribution of each text segment, thereby enhancing the overall accuracy and practicality of the system.

The baseline model we use is a non-real-time model, and such models have the advantage of analyzing more speech data during processing without considering the timing of outputs, theoretically achieving higher accuracy than real-time systems. Therefore, in our subsequent experimental design, our goal is to enhance the performance of our real-time system to match the level of this baseline model, without necessarily aiming to surpass it.

## Experimental setup

In our experimental design, we referenced the parameter settings of the real-time speaker diarization model proposed by *Coria et al. (2021)*, setting both the forward time interval (step) and the length of each audio processing segment (duration) to 0.5 s. This configuration aims to prevent overlap and loss of audio input while balancing the relationship between processing time and system performance. Too short a processing time may lead to confusion in recognition results and increased energy consumption, while too long a duration may cause users to perceive a delay. The parameter setting we chose primarily considers the balance between accuracy and latency for real-time display and does not directly affect the final experimental outcomes. Therefore, we did not optimize this parameter further; actual applications may require adjustments based on different users' needs.

We set the maximum number of speakers (max speaker) to a value far exceeding the actual number of speakers (*e.g.*, 20 people), equivalent to clustering without specifying the number of speakers, which is more in line with real-world scenarios.

For the choice of real-time speech recognition model, we opted for the Whisper base model. This decision is a trade-off between the increased latency from a larger model, which could impact the user experience, and the insufficient accuracy of a smaller model for most application scenarios.

### Setting the distance threshold $\delta_{new}$

In setting the optimal distance threshold $\delta_{new}$ for incremental clustering, we experimented with different values and determined the WDER for parameter settings of 0.5, 0.6, 0.7, and 0.8. Our experimental results, as shown in Fig. 3, indicate that a $\delta_{new}$ of 0.7 achieved the lowest WDER, regardless of whether the testing dataset involved two or three speakers. Therefore, we chose to set $\delta_{new}$ to 0.7 for subsequent experiments.

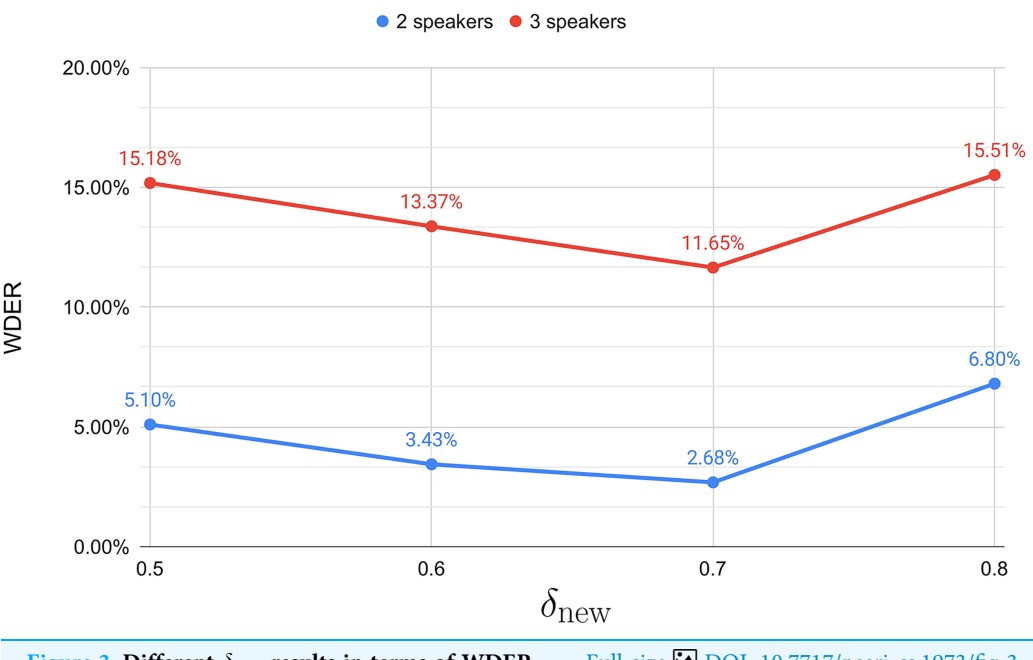

**Figure 3 Different $\delta_{new}$ results in terms of WDER.**

## Method of documenting the experiments

Finally, regarding the method of recording the real-time results, we played the audio files while running our program to record the internal audio of the computer, capturing the real-time output. We set the system to output the text and corresponding speaker label of the foremost segment when the number of segments recognized by the Whisper model reached or exceeded 3. At the end of the audio file playback, the remaining segments' corresponding speakers were recognized and recorded. Although this recording method is relatively time-consuming, considering the fluctuating nature of the Whisper model's output segments mentioned earlier and the difference in the timeline and number of segments identified by the Whisper model in real time compared to traditional file input methods, we deemed this recording approach indispensable.

## Real-time system latency testing

To ensure that the computational speed of our system would meet the requirements for real-time processing, we analyzed each audio buffer in our dataset. Specifically, we calculated the duration of the longest sentence before triggering an audio buffer pop event, as well as the time required to process this speech segment, including steps such as speech recognition, speaker embedding, and clustering. Our results indicated that the average duration of the audio buffer was 5.11 s, with a variance of 2.79; the average processing time was 0.13 s, with a variance of 0.02 s. This implies that the time our system requires to process a speech segment is only 3% of the duration of the speech segment, satisfying the conditions for real-time processing quite well and ensuring high efficiency.

## RESULTS AND DISCUSSION

Our research particularly focused on the word diarization error rate (WDER), as we deemed it a more critical metric for evaluating the performance of real-time models. Although the word error rate (WER) remains an important indicator, it was given less emphasis in this study for two main reasons: first, as mentioned earlier, we did not rigorously clean erroneous words during data processing, which could affect the accuracy and reliability of the WER; second, to ensure the low latency of the system, we chose a smaller version of the Whisper base model, which inherently limits the optimization of WER. Therefore, WDER became our primary focus for more accurately assessing the model's effectiveness in separating different speakers.

According to our results, as shown in Table 2, our model achieved an overall WDER of 6.96%, which is slightly greater than the 6.07% of the baseline model. This suggests that our real-time model maintained good stability and adaptability across various testing conditions, although it faced some challenges in more complex multispeaker interactions.

Notably, in the two-person scenario, our model showed a significant improvement in performance, with a WDER of 2.68%, which was not only lower than the baseline's 4.15% but also exceeded our expectations. This highlights our model's efficiency in handling two-speaker scenarios, which is a significant advantage for practical applications.

However, in the three-person scenario, our model's WDER increased to 11.65%, 3.72% higher than the 7.93% of the baseline model. This indicates a decrease in accuracy in complex scenarios involving more speakers, although this decline did not significantly affect the model's overall performance.

Additionally, during the experimental process, we observed that at the beginning of our system's operation, due to the nature of incremental clustering, if only a few speaker embedding features were available for analysis, the system's recognition accuracy was lower. This can lead to confusion in speaker identification during the initial stages. This situation seems to be due to a lack of sufficient data to perform effective clustering at the beginning of the system's operation.

To delve deeper into this phenomenon, we conducted an experiment aimed at observing the trend of WDER variation in the initial stages of the system, as the cumulative word count increases, in scenarios with two or three speakers. Starting from the first 100 words of the text, we calculated the WDER for every subsequent 50-word increment, continuing this calculation up to 1,000 words. The results, depicted in Fig. 4, show that for the first 100 to 1,000 words, the WDER demonstrates a decreasing trend in both the two-person and three-person scenarios. In the two-person scenario, the WDER starts at 3.82%, experiences a peak (4.84%) in the first 150 words, then declines and gradually stabilizes within the range of 2.66% to 3.25%. In contrast, the three-person scenario exhibits more pronounced initial fluctuations in WDER, starting at 24.08%, rising to 25.49%, then quickly dropping until the word count accumulates to approximately 600 words, stabilizing at approximately 14% (14.10–13.83%).

**Table 2 WDER and WER results for the baseline and our real-time system.**

| | WDER (%) | | | WER (%) | | |
|---|---|---|---|---|---|---|
| | Total | Two speakers | Three speakers | Total | Two speakers | Three speakers |
| Baseline | 6.03 | 4.15 | 7.93 | 17.46 | 15.84 | 19.08 |
| Real-time | 6.96 | 2.68 | 11.65 | 23.61 | 21.49 | 25.90 |

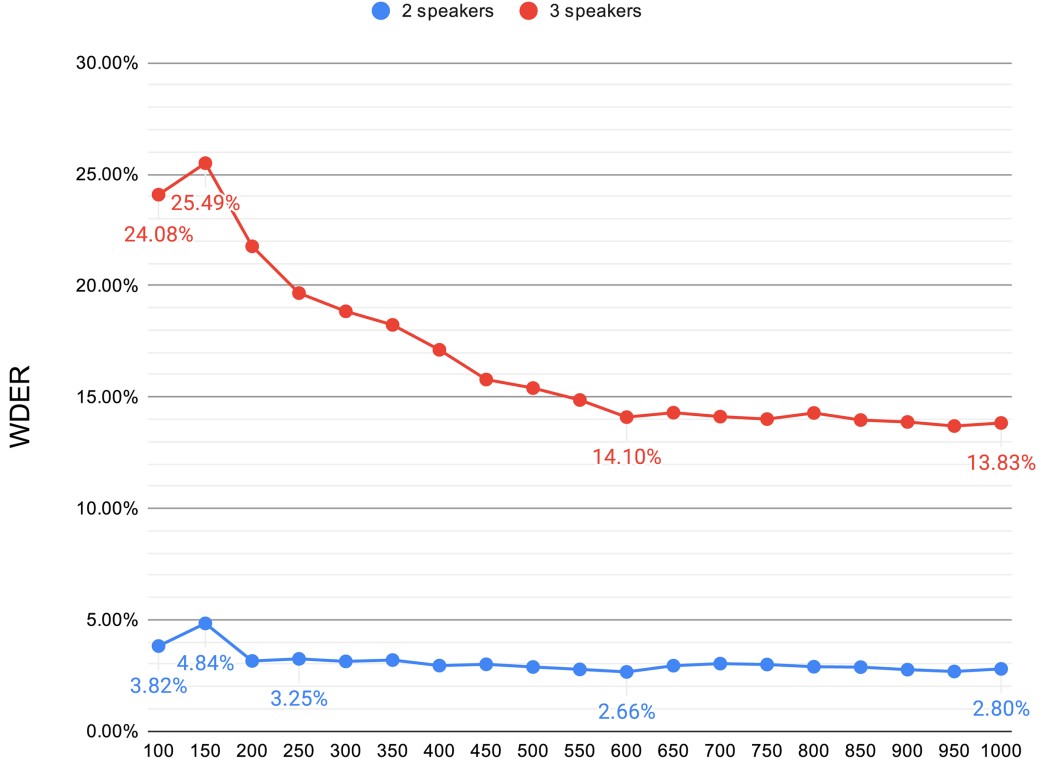

**Figure 4 Variation in the WDER within the first 1,000 words of the real-time system output.**

This analysis indicates that in scenarios involving two-person interactions, the system's recognition capabilities quickly reach stability, whereas in three-person interactions, the initial variability of the WDER is greater, and it takes longer to achieve stability. Overall, as the system processes more speaker feature data, the ability to distinguish between different speakers significantly improves, especially in scenarios involving multiple interactions, where this improvement is particularly pronounced. This further confirms that the system's recognition capabilities gradually increase as it receives more speaker feature data.

Thus, our system is more suited to scenarios where one person speaks for a period before switching to another speaker, such as in conference presentations, interview

programs, or medical consultations. In these settings, the system has the opportunity to first acquire a certain amount of speech feature data from a single speaker, enhancing the ability to distinguish different individuals through continuous updates to the clustering centroids. In contrast, scenarios with frequent speaker alternations at the onset of system operation may result in lower accuracy.

In practical applications, for the setting of the maximum number of speakers, in addition to setting it far above the actual number of participants as in our experiments, another approach is to set it to a value equal to or slightly greater than the actual number of speakers. The advantage of this setting is that if the system can correctly assign a centroid to each different speaker initially, then for subsequent speech signal inputs, the system can directly assign speakers based on the closest centroid without considering the need to add new speakers when exceeding the set threshold. However, if the system erroneously assigns the features of the same speaker to more than one centroid in the initial phase, then upon reaching the maximum number of speakers, new speaker feature inputs will not be able to increase the number of speakers and will be allocated to the closest centroid.

Furthermore, we identified another potential error pattern: if the system erroneously assigns the voice features of different speakers to the same centroid at the initial stage, this will lead to subsequent centroid updates (*i.e.*, the averaging process of features) gradually merging these inherently different speaker features. This phenomenon significantly negatively impacts the subsequent allocation of speakers, resulting in entirely incorrect assignment outcomes. Therefore, identifying and correcting such errors will be an important research direction for future system improvements.

## CONCLUSIONS

In our research, by innovatively leveraging the characteristics of the Whisper model, where each speech segment contains only a single speaker, we successfully developed a real-time multilingual speech recognition and speaker diarization system. Our system performed exceptionally well in two-person interactions, with a significant reduction in WDER from 4.15% to 2.68%, surpassing the baseline model. However, the accuracy decreased in more complex scenarios involving multiple speakers, highlighting the need for performance improvements in such situations. Nevertheless, in terms of overall performance, our system maintained a level close to that of non-real-time models even in real-time operation mode (6.03% *vs.* 6.96%). This result is significant for real-time speech processing applications, especially in scenarios requiring rapid response, such as live streaming, multiperson meetings, or multilingual customer service systems.

Overall, our study demonstrates the feasibility of using the Whisper model's segmentation for real-time multilingual speech recognition and speaker diarization systems, thereby enhancing the practicality of such systems in real-world applications. Future work will focus on further optimizing the system, particularly in complex scenarios with more frequent speaker switches, to achieve broader applications.

### Funding

This research was financially supported by the National Science and Technology Council under grant numbers 112-2410-H-182-026-MY2, as well as by Chang Gung Memorial Hospital through grant numbers NERPD4N0231 and BMRPA07. The funders had no role in study design, data collection and analysis, decision to publish, or preparation of the manuscript.

### Grant Disclosures

The following grant information was disclosed by the authors:
National Science and Technology Council: 112-2410-H-182-026-MY2.
Chang Gung Memorial Hospital: NERPD4N0231 and BMRPA07.

### Competing Interests

The authors declare that they have no competing interests.

### Author Contributions

- Ke-Ming Lyu conceived and designed the experiments, performed the experiments, analyzed the data, performed the computation work, prepared figures and/or tables, authored or reviewed drafts of the article, and approved the final draft.
- Ren-yuan Lyu conceived and designed the experiments, performed the computation work, authored or reviewed drafts of the article, and approved the final draft.
- Hsien-Tsung Chang conceived and designed the experiments, authored or reviewed drafts of the article, and approved the final draft.

### Data Availability

The models are available at GitHub:
- https://github.com/m-bain/whisperX.

### Supplemental Information

Supplemental information for this article can be found online at http://dx.doi.org/10.7717/peerj-cs.1973#supplemental-information.

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
