# Peer review of "Real-time multilingual speech recognition and speaker diarization system based on Whisper segmentation"

_PeerJ Computer Science, doi:10.7717/peerj-cs.1973_

## Round 0.1 · original submission · Major Revisions

The authors should revise the article in view of the comments and provide detailed response letter in revised submission.

**Language Note:** The review process has identified that the English language must be improved. PeerJ can provide language editing services - please contact us at copyediting@peerj.com for pricing (be sure to provide your manuscript number and title). Alternatively, you should make your own arrangements to improve the language quality and provide details in your response letter. – PeerJ Staff

·

Basic reporting

Introduction:

While mentioning the key contributions of the proposals authors haven't clearly explained which research gaps do they fill and how. E.g plz explain clearly why the existing realtime model needs improvement.

Incremental Clustering (Method)

The authors improve an existing incremental clustering approach.
Here again it will be good to elaborate which aspects of the existing method does the proposed approach improve.

Results and discussion:
The authors mention that their method is more suitable for practical realtime applications but never elaborate how.

Experimental design

Introduction:
Similar to the reporting issue, while mentioning their main contributions the authors highlight that their method is more suitable for practical realtime applications but never follow up with any quantified computations to support the claim.

Experimental Setup (Experimental Setup)
The authors manually set a lot of threshold values in their method without providing any formal validation or motivation for the values chosen. E.g
a. Advance step duration
b. Probability value for identifying silence
c. Similarity measure value for new speaker


Results and Discussion:
The authors acknowledge that their method is suitable for speakers being less than three. Can they list application scenarios for this constraint being valid? Specially considering that method is titled being for speaker diarization

Validity of the findings

Baseline(experimental setup)
Authors must justify with reasoning for selecting the baseline method they adopted.

Results and Discussion:
The proposed method achieves better results than baseline just for 2 speakers and worse for speakers bring greater than 2. This i think is a serious concern and authors must explain clearly why still their method is relevant.

Reviewer 2 ·

Basic reporting

In abstract ,, authors stated that previous schemes often fall short in speech recognition system. How authors claimed this? Is there any previous work that authors have dicussed anywhere in introduction or related work?
Most of the words seems to be AI generated in the introduction, Do authors solely used it for language purposes? If yes, there should be a decalartaion statement at the end of the paper. Moreover, i suggest to use some simple vocabulary , so the novice reader could see insights about the novelty of this work.
There is no related work, how readers will distinguish the prosord work with recent benchmarks?
Authors did'nt provide any insights about dataset classes , variations, missing values or other details that are crucial in the reproducability of this work.
How whisper model is superior to recent models of speech recognition? A clear discussion related to the superiority of proposed framework is missing.
In proposed whisper model,, authors used 1D convolutional network, but there is no clear discription of its applicability, layers, dropout, neurons, channels and other insights.

Experimental design

Needs clear decription with more details, such as dataset classes, neural network insights etc.

Validity of the findings

Results seems to be valid.

Reviewer 3 ·

Basic reporting

The authors in this paper titled “Real-time multilingual speech recognition and speaker diarization system based on Whisper segmentation” attempts to develop a real-time multilingual speech recognition and speaker diarization system, leveraging OpenAI's Whisper model. The authors are suggested to address the following comments while revising the paper.


The literature review conducted in this paper is not sufficient. The authors are suggested to add more literature on the speech recognition in general (English, Mandarin, Urdu, Arabic etc) and then they can provide a more focused literature on Mandarin with accent. It will be interesting to see how accent is being studied for other languages.

Experimental design

The title reflects that the work done is multilingual. What about the Generalization to Other Accents and Languages. The focus is on Mandarin speech with Taiwanese accents, it would be valuable to assess the generalization of the model to other languages and accents and also present and discuss the results.

Validity of the findings

a. Results and discussion need to be elaborated in more detail, also where possible compared with the existing studies.
b. Mention the weakness/limitations of this study.

Additional comments

The author needs to have a careful review before review submission the manuscript.

---

## Round 0.2 · accepted · Accept

I confirm that the authors have addressed all of the reviewers' comments. The reviewer reports for 2nd round indicate the same.

·

Basic reporting

Ok. My concerns have been addressed

Experimental design

My concerns have been addressed

Validity of the findings

The authors have addressed my concerns

Reviewer 2 ·

Basic reporting

The authors have revised the manuscript according to the previous comments. I am grateful to the authors for carefully addressing each comment and am happy to suggest the acceptance of the article.

Experimental design

No further comments

Validity of the findings

No further comments

Reviewer 3 ·

Basic reporting

The Authors have successfully addressed the comments mentioned earlier.

Experimental design

No More changes

Validity of the findings

No changes suggessted more

Additional comments

No More changes are required